# Implementation of Interventions and Policies on Opioids and Awareness of Opioid-Related Harms in Canada: A Multistage Mixed Methods Descriptive Study

**DOI:** 10.3390/ijerph19095122

**Published:** 2022-04-22

**Authors:** Camille Goyer, Genaro Castillon, Yola Moride

**Affiliations:** 1Faculty of Pharmacy, Université de Montréal, Montreal, QC H3C 3J7, Canada; camille.goyer@umontreal.ca; 2YolaRX Consultants, Montreal, QC H3W 1Y7, Canada; genaro.castillon@yolarx.com; 3Center for Pharmacoepidemiology and Treatment Science, Institute for Health, Health Care Policy, and Aging Research, Rutgers, The State University of New Jersey, New Brunswick, NJ 08854, USA

**Keywords:** risk minimization measures, policies, evaluation framework, opioid-related harms, opioid use disorders, opioids, spontaneous reporting, social media

## Abstract

In Canada, interventions and policies have been implemented to minimize the risk of opioid-related harms. This mixed methods study aimed at describing trends over time in implementation, as well as in awareness and health outcomes. For implementation, we conducted a scoping review to identify opioids interventions and policies implemented in Canada between 1 January 2016 and 15 November 2019. Awareness was measured through a descriptive analysis of opioid-related harm cases reported by consumers and health care professionals (HCPs) to the national spontaneous reporting system and of social media coverage, while health outcome consisted of opioid-related deaths recorded in the coroner’s reports database of the province of Quebec, Canada. Trends over time in implementation of interventions were compared to trends in awareness and opioid-related deaths, without implying causality. There were 413 national or provincial interventions on opioids implemented over the study period, with a four-fold increase in 2016. The most common (31.5%) was harm reduction strategies, such as naloxone distribution. The reporting rate of opioid-related harms ranged between 0.1 and 0.2 per 100,000 persons with no observed time trend. Compared to 2015, the number of social media posts increased in 2016 by 35.4% (Reddit), 329.0% (Facebook), and 381.5% (Twitter). Between 2016 and 2019, there was a slight decrease in the number of opioid-related deaths recorded in the coroner’s database. Overall, the increase in the number of policies did not see a parallel increase in spontaneous reports of opioid-related harms as an indicator of consumer or HCP awareness. Conversely, the dramatic increase in social media coverage was consistent with heightened public awareness. Although no inferences of causality were made in this study, the decrease in opioid-related deaths observed in the recent years may indicate a potential effectiveness of interventions and policies.

## 1. Introduction

Canada is in the midst of an expanding opioid epidemic. Between January 2016 and March 2019, there were 12,800 apparent opioid-related deaths, with approximately 85% occurring in the provinces of British Columbia, Alberta, and Ontario, and 20,700 emergency medical services in response to a suspected opioid overdose [1]. In 2016, approximately 20 million opioid prescriptions were dispensed, which is equivalent to nearly 1 prescription for every adult, making Canada the second-largest consumer of prescription opioids in the world, after the United States (US) [2]. According to the Canadian Public Health Association, opioid-related deaths between 2009 and 2014 increased 7-fold in British Columbia and 20-fold in Alberta, demonstrating that the opioid epidemic in Canada has been expanding as early as 2009 [3]. As an attempt to curb the opioid epidemic, officially declared in 2016, policies on opioids have been implemented in Canada, such as drug scheduling to increase accessibility to naloxone [4], communication to health care professionals (HCPs) [5] and/or patients, as well as interventions at the point of care (e.g., opioid stickers in pharmacies, triplicate prescriptions, prescription monitoring programs) in order to increase the awareness of opioid-related harm [6]. In parallel, targeted risk management plans and follow-up commitments by the pharmaceutical industry specifically for opioids have been integrated into the Canadian drug regulations in 2018 [7]. At the same time, opioids working groups have also been created at Health Canada and the Canadian Institute for Health Information. Several models and theoretical frameworks have been published in the literature for the evaluation of health interventions, including the ones proposed for therapeutic risk minimization [8]. Within these frameworks, components of effectiveness evaluation consist of coverage, process indicators (i.e., awareness, knowledge, and understanding of key safety messages) as well as health outcomes [9,10]. For opioids, the number of evaluation studies reported in the peer-reviewed literature has lagged far behind the number of interventions and policies that have been implemented thus far [11]. A lack of harmonized guidance on methodological frameworks for studies of effectiveness of therapeutic risk minimization interventions has recently been observed [12], which may have contributed to the fragmented evidence on the impact of opioid interventions. A common shortcoming of these evaluation studies is the lack of data sources for the measurement of effectiveness outcomes. To our knowledge, a comprehensive and concurrent description of all components of the therapeutic risk minimization framework (i.e., intervention coverage, process indicators, health outcomes) using a multistage mixed methods design has not yet been conducted, which was the rationale for this study. Each component was evaluated individually with no attempt of inferring any causal associations between the observed trends.

## 2. Materials and Methods

### 2.1. Overview

The principle of integration of this mixed methods research occurs at two levels [13]. The first is at the study design level, by integrating an exploratory multistage mixed methods framework design. This design was sought to focus on a qualitative description of interventions and policies that aimed at decreasing opioid-related harms over time, and to assess, in parallel, process outcomes (HCP and consumer awareness of opioid-related harms through social media and spontaneous reporting data) as well as health outcomes (opioid-related deaths in the province of Quebec, representing approximately 22.3% of the Canadian population). The second principle is the contiguous approach to integration, which involves interpreting and reporting the findings independently for each component of the evaluation framework, as described below.

The following components of an evaluation framework were addressed in our study: (a) Implementation of risk minimization interventions and policies on opioids in Canada; (b) Process measures consisting of awareness measured by the reporting of opioid-related harms by HCPs and consumers to the national spontaneous reporting system (Canada Vigilance) as well as social media monitoring; and (c) Health outcome, measured by opioid-related deaths recorded in a provincial Coroner’s report database (Quebec only). These measures were selected on the basis of their availability at the national or, alternatively, the provincial levels. Trends over time in the implementation of interventions were compared to trends in awareness and in opioid-related deaths. These outcomes could not be used for hypothesis-testing purposes regarding the effectiveness of interventions, since they could not be linked to one another at the patient or HCP level. Hence, for this project, each component has been individually analyzed using a multistage mixed methods approach with an ecological perspective, without implying causality.

### 2.2. Implementation: Scoping Review of Interventions and Policies

A scoping review was conducted, involving a literature search of Canadian interventions (i.e., risk mitigation strategies, education and/or continuing education, public opioid awareness programs, policies, update or creation of guidelines/standard of practice, knowledge exchange, pharmacovigilance, control, and monitoring, community interventions, prevention measures) aiming to minimize or mitigate opioid-related harms, or increase awareness of the opioid crisis. The search was conducted using MEDLINE and Embase over the period from 1 January 2016 through 15 November 2019 (date last searched). The year 2016 was selected as it corresponded to the year when the opioid crisis was officially declared by the government of Canada [14]. The search strategy is presented in Appendix A. There was no restriction on the geographical scope for the search strategy, as relevant sources may not be indexed using the country as a keyword. National interventions were those implemented at a federal level (e.g., by Health Canada, the national regulatory agency) whereas provincial interventions were those implemented by provincial governments (according to the Canada Health Act, the provision of health care services is under the responsibility of provinces) [15]. For additional sources, pragmatic searches of relevant websites of governmental and non-governmental organizations were also conducted in English and French using Google (Accessed on 13 November 2019: www.google.com/) and Google Scholar (Accessed on 13 November 2019: www.google.com/scholar/) search engines. We included publications (i.e., original studies, literature reviews, conference proceedings, opinions, editorials) that described interventions and policies aiming to minimize or mitigate opioid-related harm in Canada, implemented between 1 January 2016 and 31 December 2019, and excluded those that did not explicitly mention implementation in Canada or in one of its provinces.

For each source retained in the review, the following data were extracted: Type of intervention (i.e., policy change, educational material, awareness material, update of treatment guideline or creation of a treatment guideline, pharmacovigilance, control, and monitoring), name, date of first implementation in Canada and/or date the intervention was announced on the website (expressed in quarter and year) during the study period, and whether it had provincial or national coverage. If only the year was reported, the date of implementation was assigned as the first quarter (Q1) of the year. Publications describing interventions aiming to minimize opioid-related harms were retained while those that did not mention interventions or mentioned interventions outside of Canada were excluded. A qualitative description of interventions was conducted and using the date of implementation, a graphical representation of the number of interventions over time was derived along with a visual assessment of trends.

### 2.3. Awareness Outcome: Spontaneous Reports of Opioid-Related Harms

Canada Vigilance is the national post-market surveillance program that collects spontaneous reports of suspected adverse effects of prescription and non-prescription health products made by HCPs or consumers in Canada [16]. It does not cover the cases associated with illicit opioid usage. Although a useful tool for drug safety signal detection at the country level, spontaneous reporting data are associated with well-known limitations for the estimation of incidence of adverse events, since reporting is greatly influenced by external factors such as media coverage, regulatory communications, time since product market launch, etc. [17,18]. However, being highly influenced by those external factors, increases in reporting are known to be the result of increased awareness of identified or suspected risks [17,18]. Hence, we considered trends in spontaneous reporting as an indicator of awareness of opioid-related harms.

We included all case reports of overdose, opioid-related death, abuse, misuse, dependence, and diversion with an opioid as the suspected drug between 1 January 2009 and 31 August 2019 (last date available). The year 2009 was selected as a starting point based on the literature, reporting that the opioid epidemic in Canada has been expanding as early as 2009. Opioid-related harms were first identified using predefined Preferred Terms (PTs) in the Medical Dictionary for Regulatory Activities (MedDRA, Version 22.1). We excluded case reports that mentioned opioids as a concomitant drug (not the suspected drug), those that were intentional adverse drug reactions (ADRs) (e.g., suicide), cases with missing data on the product information (not clear if a harmful event was suspected to have been caused by an opioid), and invalid case reports (i.e., lack of an identifiable patient, reporter, adverse reaction term or health product) [9].

Descriptive analyses were conducted on the following patient characteristics: Age group (Neonate: 0–<25 days, Infant: >25 days–<1 year, Child: ≥1–<13, Adolescent: ≥13–<18, Adult: ≥18–65, Elderly: >65), sex, type of opioid-related harms including seriousness (based on the regulatory criteria) [19], suspected opioid(s) (i.e., opioid name, route of administration, dosage form (short acting, long-acting, extended release)), reporter type (HCPs or non-HCPs/consumers), and number of reports by quarter. In the absence of data on the total number of prescription opioids users in Canada, the reporting rate was estimated by using the absolute number of spontaneous reports per quarter as the numerator and the quarterly Canadian population estimate, derived from Statistic Canada [20], as the denominator.

### 2.4. Awareness Outcome: Social Media Posts

Information on the public level of awareness of the opioid crisis and of the implementation of interventions and policies, was sought through digital surveillance of Reddit, Twitter, and Facebook over the period 2009–2019. These generic social media networks were used given their free public access. Hence, specialized health care social networks or forums (e.g., Drugs-Forum.com, accessed on 20 November 2019) that did not allow public access were excluded. Posts on recreational opioid use were also excluded.

For Reddit, a program was developed using the website’s application programming interface (API) to extract relevant posts. Duplicate posts were removed by cross-checking the post ID provided in the extracted file. For Twitter, Twitter-scraper [21], known as a web scraper program, was used to extract posts with the topic and geographical scope of interest. This program extracts structured data from Twitter posts within a time frame and by using specific keywords. For Reddit and Twitter, the program extracted structured data by selecting posts with the following keywords: ‘Canada’ and ‘Opioid(s), including their specific names’ or ‘Opioid-related harm, including other keywords associated with opioid use’. Duplicates were removed by the ‘tweet Id’ and the list of posts was then manually screened to identify unwanted posts (e.g., drug advertising). For Facebook, Canadian public groups referring to substance use, drug use, and the Canadian opioid crisis (e.g., Save Our Young Adults from Prescription Drug Abuse) were identified before initiating data extraction. These groups were identified using the keywords: Drugs, Canada drugs, substance use, and opioid. Each group was then analyzed to determine whether they were Canadian public groups referencing the opioid crisis. For feasibility reasons, a manual screening of public posts in each group was conducted since Facebook no longer allows for programs to scrape groups or profiles.

For each post retained, the following general information was extracted: Website (i.e., Reddit, Twitter, Facebook) and the post date (quarter and year). Frequencies and percentages were obtained for categorical variables (website name, group names (Facebook groups, subreddits), source publication) while the number of posts, minimum, and maximum were calculated for each time point.

### 2.5. Health Outcome: Opioid-Related Death (Province of Quebec Only)

All deaths that are accidental, intentional, homicidal, or unknown cause of opioid overdoses are brought to the coroner’s office for analysis. Coroner’s databases are maintained individually for each province, and, for this project, public access was only available for the province of Quebec covering the period from 1 January 2009 through 31 December 2017 (last date available). The opioid-related death cases were identified using the International Classification of Disease, 10th Revision (ICD-10) codes: T40.X (Poisoning by, adverse effect of and underdosing of narcotics and psychodysleptics [hallucinogens]) referring to an opioid-overdose leading to death (T40.2 (other opioids), T40.4 (other synthetic narcotics), T40.1 (heroin), T40.3 (methadone), T40.6 (narcotics), T40.0 (opium)). The type of opioid(s) suspected of causing the death was identified through an examination of the report narratives.

Frequency distribution was obtained for age group at the time of death (Neonate: 0–<25 days, Infant: >25 days–<1 year, Child: ≥1–<13, Adolescent: ≥13–<18, Adult: ≥18–65, Elderly: >65), sex, and the suspected opioid(s). Acquisition channel (prescription or illicit) was not available.

A graphical representation of the number and characteristics of opioid-related deaths by year was derived, and a visual analysis of trends was undertaken.

## 3. Results

### 3.1. Implementation: Scoping Review of Interventions and Policies

A total of 413 interventions and policies aiming to reduce opioid-related harms or increase awareness of the opioid crisis were identified in Canada between 2016 and 2019: 129 (31.2%) were national (i.e., implemented in all provinces, including Quebec) while 284 (68.8%) were provincial (implemented in one province only) (Appendix A). The number of province-specific interventions varied greatly, with British Columbia having the most (n = 102; 24.7%), followed by Ontario (n = 80; 19.4%), Alberta (n = 37; 9.0%), and Quebec (n = 20; 4.8%). National interventions, implemented in all provinces, also increased over time. Trends over time in the number of interventions by province is shown in Figure 1. Overall, there was an increase in the number of interventions over time, with a plateau starting 2019 Q1. From 2016 (reference period, n = 99 newly implemented interventions and policies) to 2019 (n = 413 interventions and policies, cumulative at the national or provincial level), there was a 4.2-fold increase in the number of interventions and policies across all provinces.

As shown in Table 1, the most common types of interventions were harm reduction or risk mitigation strategies (n = 130; 31.5%), followed by education and/or continuing education (n = 101; 24.5%), public opioid awareness programs (n = 65, 15.7%), and policies (n = 51, 12.3%). Examples of harm-reduction strategies include Take-Home Naloxone programs across the country [22], drug checking services for fentanyl for drug users [23], and centers for substance use [24]. Of the 101 education strategies identified, 71 (70.3%) targeted HCPs, 24 (23.8%) were community-based (patients and non-patients), and 6 (5.9%) targeted patients. Examples of education and/or continuing education for HCPs are: training courses specific to opioids, improving prescription practices, and treating opioid-related harms. Other material consisted of webinars and workshops on how to detect an opioid overdose [25] and on how to use a naloxone kit [26]. Examples of opioid awareness resources for the community are: Video (End Stigma Campaign) [27], web series on patients living with opioid use disorder [28], and fact sheets on opioids and pain management [29]. Of the 65 awareness strategies, 57 (87.7%) targeted the community (i.e., non-patients and patients) and 8 (12.3%) targeted patients only. An example of policy change was the removal of naloxone from the prescription list (Schedule I to Schedule II) in Canada, effective as of March 2016 [4]. Establishing take-home naloxone programs across Canada may increase access to naloxone for opioid users and for their friends and family before the arrival of first responders.

### 3.2. Awareness Outcome: Spontaneous Reports of Opioid-Related Harms

In Canada Vigilance, there were 6727 reports of opioid-related harms, corresponding to 4970 patients between 1 January 2009 and 31 August 2019. Patient characteristics are further described in Table 2. There was a predominance of males (n = 2730; 54.9%). The mean ± standard deviation (SD) patient age was 38.1 ± 17.0 years and, after excluding unknown age groups, adults age ≥18–<65 years accounted for the majority of opioid-related harm cases (67.5% overall and 90.0% when excluding reports of unknown age). The vast majority of reported opioid-related harms were categorized as serious (n = 4881; 98.2%). The types of opioid-related harms consisted of abuse, misuse and/or dependence (n = 3986; 59.3%), followed by opioid-related death (n = 1344; 20.0%). Oxycodone was the most frequently suspected opioid (n = 2047; 30.3%), followed by hydromorphone (n = 1402; 20.8%) (Appendix A).

Between 2009 Q1 and 2018 Q3, the reporting rate varied between 0.1 and 0.2 spontaneous reports per 100,000 persons/quarter (Appendix A). However, there were two reporting quarters that experienced peaks in reporting. The first occurred in 2012 Q1, with a reporting rate 0.5 per 100,000 persons, and the second was in 2018 Q4, when the reporting rate rose to 1.8 reported cases per 100,000 persons. The latter period corresponds to a simultaneous transmission of cumulated reports originating from marketing authorization holders (MAHs) following the implementation of the new regulation on opioids (87.4% of the total number of cases reported during that quarter) [30]. As part of Health Canada’s Opioid Action Plan, MAHs were required to develop and implement Canadian-specific risk management plans for opioid products in order to monitor, prevent, and mitigate the risks associated with the use of opioids.

### 3.3. Awareness Outcome: Social Media Posts

Over the study period, the number of posts related to the Canadian opioid crisis, or an adverse event suspected to have been caused by an opioid in Canada, was 1619 for Facebook, 1132 for Reddit, and 43,058 for Twitter. Graphical representation of trends over time in each social media can be found in Figure 2. Overall, regardless of the type of social media, the number of posts increased as of 2016 Q1. Between 2015 and 2016, there was a 35.4%, 329.0%, and 381.5% increase of posts for Reddit, Facebook, and Twitter, respectively. Trends after the year 2016 were comparable between Reddit and Twitter: the frequency of posts slightly decreased after 2016 Q4 and plateaued as of 2017 Q1 through 2019 Q3. However, on Facebook, trends continued to increase after the year 2016. There was a sharp increase of posts in 2019 Q1, which corresponds mainly to the high activity from one public group on Facebook that shared content on the number of overdoses related to an opioid.

### 3.4. Health Outcome: Opioid-Related Death (Province of Quebec Only)

In the Quebec coroner’s reports database, there were 1582 confirmed opioid-related deaths between 1 January 2009 and 31 December 2019.

The distribution of death cases by age group and sex is reported in Table 3. Mean age ± SD was 44.8 ± 13.3 years and most cases involved males (n = 1074; 67.9%). Hydromorphone was the leading opioid causing death (n = 557; 22.9%), followed by morphine (n = 425; 17.5%), and fentanyl and its derivatives (acetyl-fentanyl, butyryl fentanyl, carfentanyl, fluorobutyryl fentanyl, furanyl-fentanyl, norfentanyl, and par-fluorobutyryl fentanyl) (n = 323; 13.3%) (Appendix A).

As shown in Figure 3, the annual number of opioid-related deaths in Quebec increased from a total of 108 cases in 2009 to 147 in 2019, although a reversal in trend was observed starting in 2017. The number of opioid-related deaths associated with fentanyl and its derivatives increased by 6.8-fold between 2009 Q1 and 2017 Q4 (Appendix A) and decreased from 75 deaths in 2017 to 25 deaths in 2019. The relative distribution of fentanyl- and oxycodone-related deaths to the total number of deaths between 2016 and 2019 slightly decreased over time (respectively, 18.9% and 18.0% of deaths in 2016, and 12.7% and 10.2% of deaths in 2019). Methadone-related deaths increased between 2009 and 2019 (respectively, 5.3% and 13.7% of deaths). Graphical representations of the relative distribution of fentanyl-, oxycodone-, and methadone-related deaths may be found in Appendix A.

## 4. Discussion

This mixed methods study provided a description of trends over time in the number and type of opioid interventions and policies implemented in Canada since the start of the opioid epidemic in 2016, as well as in the reporting of opioid-related harms and social media coverage (awareness), and in the number of deaths due to opioid overdose in Quebec (health outcome). Collectively, measures that have been used cover all indicators of the framework for the evaluation of effectiveness of therapeutic risk minimization interventions [10]. Findings from this study may be relevant elsewhere, as opioid-related harm has become a public health crisis in many countries globally.

From the scoping review, we observed that the geographical distribution of the interventions implemented reflected areas in Canada most affected by the opioid crisis, namely British Columbia, Alberta, and Ontario [31]. Moreover, social media coverage on the level of opioid awareness also increased over time since 2016. The relatively stable spontaneous reporting rate over time of opioid-related harms associated with prescription opioids was not suggestive of an awareness effect between 2016 and 2019 (Appendix A). Instead, a large number of simultaneous transmissions of cumulated reports originating from MAHs occurred at the end of the year 2018, due to the new regulation implemented by Health Canada, which requires pharmaceutical companies to report any opioid-related harm [32].

Our findings reinforce the fact that social media is a recognized source of data when assessing awareness of opioid-related harms in the community, as previously reported by other studies [33,34,35,36]. According to a recent review, social media is also sensitive to measure the effect of risk communication interventions, such as black box warnings and label changes [36]. A unique strategy was used in our study to measure the level of awareness of the opioid crisis by using two complementary types of data sources, i.e., spontaneous reports and social media, which has not yet been documented in the literature. Spontaneous reporting systems, such as Canada Vigilance, traditionally used for signal detection purposes, may also be a useful tool to assess the awareness in the community, especially in Canada, where there is a paucity of nationwide data sources [17,19]. Absence of time trends in AE reporting may be due to the social and sensitive nature of opioid-related harm. While AE reporting has been previously used as a proxy of awareness of the risks associated with statin usage [17], reporting practices for opioid-related harm may differ from that of other drugs. Further research would be needed to understand the determinants of reporting practices for opioid-related harm.

Many opioid-related deaths were due to strong opioids such as hydromorphone, fentanyl, methadone, and oxycodone. Between 2009 and 2017, we observed an increase in fentanyl-related deaths along with a decrease in oxycodone-related deaths, which may reflect changes in prescription practices due to the introduction of potent opioids like fentanyl, which was highlighted in a recent study [37]. Starting in 2017, there was a reversal in the trend toward a decrease in opioid-related deaths in Quebec. However, it is unclear as to why methadone-related deaths constantly increased over time. These findings suggest that the evaluation of interventions and policies on opioids should also address individual products as opposed to opioids in general.

This study does have some limitations. Interventions that were not published on the web or in the literature may have led to an underestimation of the true efforts made to reduce opioid-related harms in Canada. The total number of users of individual prescription opioid products in Canada was not available, which did not allow for the estimation of a reporting rate [38]. Spontaneous reports of opioid-related harms are only provided at a national level and thus, disaggregation of spontaneous reporting of opioid-related harms by province is not possible. The number of posts made on public social media sites without adjusting for confounding variables may not accurately represent awareness. The type of posts and what it was referring to (e.g., opioid awareness, adverse drug reaction) was not analyzed, and only the volume of posts on the opioid crisis in Canada was measured. In addition, the inclusion of posts from specific groups aiming to raise awareness on opioids/drugs may overestimate the general public’s awareness on the matter [38]. Opioid-related deaths were assessed using the coroner’s reports database, which is a robust and objective measure but available only at the provincial level, which limits the generalizability of the findings on the health outcomes. Furthermore, it was not possible to distinguish between prescribed and illicit opioid usage. Finally, study data cannot be used for hypothesis-testing purposes regarding the effectiveness of risk minimization interventions and policies, since they cannot be linked to one another at the patient or HCP level. Hence, for this project, these have been individually analyzed using a mixed methods approach under an ecological perspective, without implying causality.

## 5. Conclusions

To our knowledge, a comprehensive and concurrent description of all components of the risk minimization framework (i.e., intervention coverage, process indicators, health outcomes) using a multistage mixed methods design has not yet been conducted, which was the rationale for this study. Although no inferences of causality were made in this study, the slight decrease in opioid-related deaths in the recent years may reflect a potential effectiveness of interventions and policies. However, it is not possible to determine which interventions and policies are the most effective.

## Figures and Tables

**Figure 1 ijerph-19-05122-f001:**
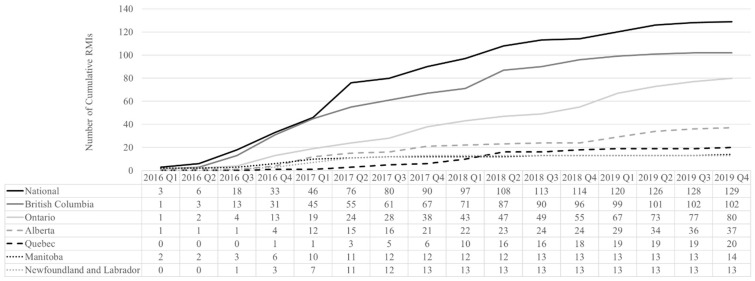
Distribution of opioid interventions and policies in Canada between 2016 and 2019 by year of implementation and province.

**Figure 2 ijerph-19-05122-f002:**
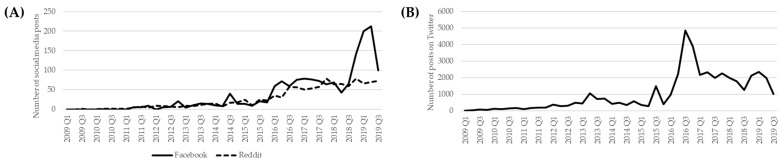
(**A**) Number of Facebook and Reddit posts mentioning the opioid crisis and/or opioid-related harms on Facebook (n = 1619) and Reddit (n = 1132) between 2009 and 2019; (**B**) Number of posts mentioning the opioid crisis and/or opioid-related harms on Twitter (n = 43,058) between 2009 and 2019.

**Figure 3 ijerph-19-05122-f003:**
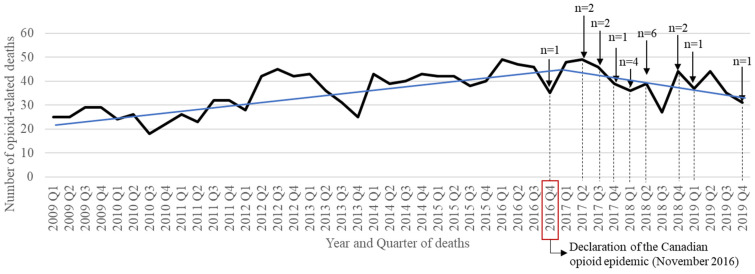
Number of opioid-related deaths in Quebec between 2009 and 2019. The arrows and corresponding dashed lines represent the number of interventions and policies implemented at that moment in time. The linear blue lines represent the trend in the number of opioid-related deaths in time before and after the declaration of the Canadian opioid epidemic.

**Table 1 ijerph-19-05122-t001:** Types of opioid risk minimization interventions and policies implemented in Canada (national or provincial) between 2016 and 2019.

Type of Interventions *	n (%) Total = 413
Harm reduction strategies	130 (31.5)
Education/continuing education	101 (24.5)
For HCPs only	71 (70.3)
For the community	26 (25.7)
For patients only	4 (4.0)
Opioid awareness	65 (15.7)
For the community	57 (87.7)
For patients or family only	8 (12.3)
Policy changes	51 (12.3)
Update/creation of guidelines/standard of practice	28 (6.8)
Knowledge exchange	16 (3.9)
Pharmacovigilance, control, and monitoring	11 (2.7)
Community interventions	9 (2.2)
Prevention measures	2 (0.5)

HCP: Health care professional. * Categories are mutually exclusive. Note: Interventions targeting patients apply to opioid users or their families (e.g., the Neighbourhood Pharmacy Association of Canada Created a handout named “Opioid Pain Medicines Information for Patients and Families”). Interventions targeting the community involve those directed towards a broader population (e.g., the University of Waterloo created a video on Naloxone administration). Interventions that target the community are also likely to reach HCPs and patients.

**Table 2 ijerph-19-05122-t002:** Characteristics of patients and reporters of opioid-related harm cases reported to Canada Vigilance between 2009 and 2019.

Patient Characteristics	n (%) Total = 4970
Sex	
Male	2730 (54.9)
Female	1969 (39.6)
Unknown	271 (5.5)
Age	
Mean ± SD, in years	38.1 ± 17.0
Neonate (0–<25 days)	8 (0.2)
Infant (>25 days–<1 year)	16 (0.3)
Child (≥1–<13 years)	44 (0.9)
Adolescent (≥13–<18 years)	76 (1.5)
Adult (≥18–<65 years)	3356 (67.5)
Elderly (≥65 years)	228 (4.6)
Unknown	1242 (25.0)
Seriousness of opioid-related harm	
Serious	4881 (98.2)
Non-serious	89 (1.8)
Type of reporter	
Non-HCP	3174 (63.9)
HCP	1501 (30.2)
Unknown	295 (6.0)
Opioid-related harm ^a^	*Total = 6727*
Abuse/Misuse/Dependence	3986 (59.3)
Opioid-related death	1344 (20.0)
Overdose	1159 (17.2)
Diversion	38 (3.5)

SD: Standard deviation, HCP: Health care professional, Non-HCP: Non-health care professional. ^a^ Some case reports reported more than one opioid-related harm.

**Table 3 ijerph-19-05122-t003:** Characteristics of patients for opioid-related deaths in Quebec (Coroner’s reports) (2009–2019).

Patient Characteristics	n (% Total = 1582
Sex	
Male	1074 (67.9)
Female	508 (32.1)
Age	
Mean ± SD, in years	44.8 ± 13.3
Pediatric (0–<13 years)	-
Adolescent (≥13–<18)	3 (0.2)
Adult (18–<65 years)	1498 (94.7)
Elderly (>65 years)	80 (5.1)
Unknown	1 (0.1)

SD: Standard deviation.

## Data Availability

The data used are not available from the investigators.

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
