# Peer review of "Implementation of Interventions and Policies on Opioids and Awareness of Opioid-Related Harms in Canada: A Multistage Mixed Methods Descriptive Study"

_ijerph, 2022, doi:10.3390/ijerph19095122_

Round 1

Reviewer 1 Report

IJERPH-1617809-v1.
England, March 2022.

The article by Goyer, Castillon and Moride is focusing on opioid-related deaths, policies and interventions in Canada. Authors give substantial evidence on how these are intertwined together and influencing each other. Substantial amount of statistics are presented, including several time-courses, both in the main manuscript and as supplemental figures. The article is well written and well designed. These provide ease of read for potential readers. Many appraisals can be given to the current form of the manuscript. However, author may want to consider the following few points below (very minor):

-Throughout the manuscript : please use italics when referring to Latin idiomatics, such as id est (ie) and et cetera (etc…).

-Line 59, references 10 and 11 should be combined towards: [10-11].

-Line 159 seems to contain a double space between “quarter” and “In the absence”.

-Lines 215, 219, 228. I am unsure what “Error! Reference source not found” means.

-Figure 1, line 224. Here, the distinction between “British Columbia” and “Manitoba”, both displayed as solid grey lines, should be re-designed. Similarly, “National” and “New Foundland and Labrador” are both displayed as solid black lines. Please edit to allow for easy distinctions between Provinces. After zooming (excessively) on the key, I believe the dashes are not easily distinguished and were truncated. Authors should consider editing this Figure.

-Lines 243-244. This is a very interesting information. Authors should expand more and explain in further details how this would affect the public/legislation.

-Lines 252-253. Authors should be careful here when stating that “67.5% of cases were adults”, since 25% of these are in “unknown” age groups. Here, I suggest that authors could maybe rephrase that sentence towards: “in cases of opioid use from known age groups, adults represented the majority…”, or even “after excluding unknown age groups, adults were found to be the majority of opioid-related harm cases…”.

-As a general comment, after reading lines 312-313, I believe that the word “epidemic” could be rephrased to “pandemic”. Indeed, Canada is far from being the only country where opioids are reaching pandemic levels. Out of interest, authors might want to discuss this (discussion section?). Please see the following articles:

doi:10.1001/jama.2019.20844

10.1016/S0140-6736(18)31073-0

In its current form, the manuscript is almost ready for publication. I therefore recommend acceptance following minor editing. Bravo à vous trois.

Author Response

Please see the document Word uploaded for our answers to your comments. 

Reviewer 2 Report

Goyer et al report a mixed methods study aimed describing trends over time evaluating Interventions and policies implemented in Canada to minimize the risk of opioid-related harms between 1st January 13 2016 and 15th November 2019.

Authors have made an important effort to collect all the data presented in the manuscript. However, I have a mixed feeling because the relevant limitations for connecting the data sets presented. Although the authors clearly limit the potential inference of the collected data, it is difficult not to do that. Each set of data have value itself but their relationship is very questionable and the authors should stress this in the discussion and conclusions

I would like to make some comments (some already mentioned by the authors) that make difficult to draw non-biased conclusions from the work done:

  • It is no clear to me if reports to the national spontaneous reporting system are valid proxy and are truly related to awareness of risks associated to opioids consumption. Given the characteristics of opioids consumption (even commercial opioids are subject to “illegal” use) it is possible that consumers are less prone to report AEs than with other therapeutic drugs. This would be valid for both consumers and HCP.
  • Awareness evaluation through social media is subject to bias:
    • Could include recreational drug use.
    • Inclusion of groups could overestimate the general public awareness. It is well known that some groups are very active in social media and do not represent the whole population.
  • Spontaneous reports of opioid-related harms are only provided at national level. Disaggregation by province level should be needed.
  • I think that Graphical representation of trends of hits in social media should be included in the main text of the manuscript instead in supplemental material.
  • Interventions are very different among provinces (only 5% in Quebec). So, it is not possible to relate the interventions national figures to Quebec opioids related deaths.
  • Death rates in Quebec could not represent those in other provinces. May be the low number of interventions in Quebec make a difference. I wonder if there is any relationship between the intervention and deaths in Quebec (may be with some lag-time). Although no causality could be concluded it would be interesting.
  • It is no clear to me the conclusion on “the continued rise in opioid-related deaths”. According to Fig 2 the number of deaths is quite stable in Quebec since 3th Q2012; even a sligh decrease can be observed from 2Q2017 (probably no very significant).

Author Response

Please see the document attached for our answers to your comments. 

Reviewer 3 Report

In the current study, the authors have aimed at describing trends over time in implementation of interventions, awareness, and health outcomes. Authors have gathered and compiled the data for the respective attributes. However, there are no significant conclusions that can be drawn from the study. 

Although the article is well structured into section and subsections, and it is within the scope of journal. There are some major concerns that needs to be addressed to improve the article:

  • The title of the article “Evaluation of effectiveness of interventions and policies on opioids in Canada: A multistage mixed methods descriptive study”, is not justified by the study conducted.

A total of 413 interventions and policies implemented at national and provincial level between 2016 and 2019 are mentioned, while Quebec has only 9 implemented interventions. It would be more informative if the impact of the interventions in Quebec at least were discussed considering the opioid related deaths in Quebec region. Since authors mentioned the unavailability of these data from other regions.

  • The conclusion section needs to be rewritten. The conclusions are not well supported by the data presented. The rationale for conducting the study also needs to be clearly presented.

Based on figure 5, the number of opioids-related deaths in Quebec region exhibit a decreasing trend and not the increasing trend, as mentioned in the text.

Some minor comments include:

  • Page 5, line 215, 219, 228 – There are typos, “Error! Reference source not found..”
  • Page 6, Table1: The table layout is not very clear, currently it gives impression that there are duplicate data entries – “For the community” and “For patients”.
  • The referencing format needs to be consistent. For instance, check reference 8 and 16.
  • Another suggestion would be to include the data points till 2021. This will provide the updated information.

Author Response

Please see the attached file for our answers to your comments. Thank you.

Reviewer 4 Report

This study is focused on the interventions and policies that have been implemented in Canada to minimize the risk of opioid-related harms. It accurately describes trends over time in implementation, awareness outcome, both as spontaneous reports of opioid-related harms and social media posts, and health outcomes in the province of Quebec.

This is a very accurate study, in which each section is properly presented and illustrated. The huge amount of data collected and analyzed has originated numerous tables and figures, each of which is clearly described and discussed.

Despite some limitations that have been recognized by the authors, this study is highly informative and provide statistical evidence, among others, that between 2009 and 2019  the number of opioid-related deaths in Quebec has not decreased over time, that many opioid-related deaths were due to strong opioid agonists and that the number of fentanyl-related deaths increased while that of oxycodone-related deaths decreased over time. The design of the study is clear, the search of the sources well described and the statistical analysis properly performed.

I have only minor suggestions for this nice study.  Tables 2 and 4, illustrating the specific opioids that caused deaths and opioid-related harms should be grouped in a unique Table and shifted from Suppl. Materials to the article. Similarly, information about the patient characteristics of the opioid-related deaths, i.e. Table 3 of Supll Mat., are important and should incorporated into the main article.

Author Response

(The authors gave the same response as above.)

Round 2

Reviewer 2 Report

The authors answer satisfactorily my previous comments. I think that the manuscript is now acceptable for publication.

Best,

Reviewer 3 Report

Authors have addressed all the raised concerns and diligently revised the manuscript. The manuscript in the current form is suitable for publication.